# Recognition Performance Analysis of a Multimodal Biometric System Based on the Fusion of 3D Ultrasound Hand-Geometry and Palmprint

**DOI:** 10.3390/s23073653

**Published:** 2023-03-31

**Authors:** Monica Micucci, Antonio Iula

**Affiliations:** School of Engineering, University of Basilicata, 85100 Potenza, Italy; monica.micucci@unibas.it

**Keywords:** multimodal systems, palmprint, hand-geometry, 3D ultrasound, fusion

## Abstract

Multimodal biometric systems are often used in a wide variety of applications where high security is required. Such systems show several merits in terms of universality and recognition rate compared to unimodal systems. Among several acquisition technologies, ultrasound bears great potential in high secure access applications because it allows the acquisition of 3D information about the human body and is able to verify liveness of the sample. In this work, recognition performances of a multimodal system obtained by fusing palmprint and hand-geometry 3D features, which are extracted from the same collected volumetric image, are extensively evaluated. Several fusion techniques based on the weighted score sum rule and on a wide variety of possible combinations of palmprint and hand geometry scores are experimented with. Recognition performances of the various methods are evaluated and compared through verification and identification experiments carried out on a homemade database employed in previous works. Verification results demonstrated that the fusion, in most cases, produces a noticeable improvement compared to unimodal systems: an EER value of 0.06% is achieved in at least five cases against values of 1.18% and 0.63% obtained in the best case for unimodal palmprint and hand geometry, respectively. The analysis also revealed that the best fusion results do not include any combination between the best scores of unimodal characteristics. Identification experiments, carried out for the methods that provided the best verification results, consistently demonstrated an identification rate of 100%, against 98% and 91% obtained in the best case for unimodal palmprint and hand geometry, respectively.

## 1. Introduction

In recent years, biometric recognition is acquiring increasing popularity in various fields where personal security is required, replacing classical authentication methods based on PINs and passwords. Biometric characteristics are mainly employed in commercial applications such as smartphones and access control, government, and forensics.

Biometric systems based on the combination of two or more characteristics, referred to as multimodal systems, have several advantages compared to their unimodal counterparts as they allow improved recognition rate, universality, and the authentication of users for which one of the single biometric characteristic cannot be detected [1,2,3]. In particular, multimodal systems based on a single sensor are arousing interest because they permit to achieve cost-effectiveness and improved acceptability from users [4].

Multimodal systems are often employed for human hand characteristics including hand geometry and palmprint because both are universal, invariant, acceptable and collectable [5,6].

Over the years, several technologies have been experimented with for the acquisition of the two human hand modalities. The most commonly employed are optical and infrared [7]. The former is mainly based on CCD cameras and contactless technique [8,9,10]: CCD cameras collect high-quality images but are limited by the bulkiness of the device, while contactless modality is highly useful for acceptability of users and reasons of personal hygiene, but it is not very reliable for low-quality images. Regarding the latter, both Near-Infrared (NIR) and Far-Infrared (FIR) radiation are used [11,12]. The principal limit of these technologies is their capability of providing information present only on the external skin surface.

Ultrasound is a technology employed in several fields including sonar [13], motors and actuators [14], Non-Destructive Evaluations (NDE) [15], Indoor Positioning Systems (IPS) [16], medical imaging [17] and therapy [18], and biometric systems [19]. The capability of ultrasound to penetrate the human body can be very useful in the latter field because it allows for 3D information on the features to be obtained, leading to a more accurate description of the biometric characteristic and hence, improved recognition accuracy [10].Moreover, ultrasound is featured by the capability of effectively detecting liveness during the acquisition phase, by simply checking vein pulsing, making the system very difficult to counterfeit and is not influenced by the presence of oil and ink stains on the skin and by environmental changes in light or temperature. Ultrasound technology has been widely investigated in the biometric field, particularly for extraction of fingerprint features [20,21] and, recently, the integration of the sensor in smartphone devices became reality [22]. Other characteristics, including hand geometry [23,24], palmprint [25,26,27], and hand veins [28,29,30] were also investigated.

In a recent paper [31], a single-sensor multimodal system based on the combination of ultrasound hand geometry and palmprint was proposed. The 3D features for both hand geometry and palmprint were extracted from the same volumetric hand images. Verification and identification experiments were first performed by separately considering the single modalities and, then, a preliminary attempt of fusion was performed by considering only the best scores of the two characteristics.

In the present work, the abovementioned study has been extensively carried out by testing several fusion approaches based on the weighted score sum rule and by considering a wide variety of possible combinations of palmprint and hand-geometry scores.

The remainder of this paper is structured as follows. The main papers related to the present work are reviewed in Section 2. In Section 3, the acquisition modality of hand volume and feature extraction techniques is described. Section 4 focuses briefly on state-of-the-art fusion techniques and a description of experimental methods. In Section 5, fusion results obtained via verification and identification experiments are reported. Lastly, concluding remarks are provided in Section 6.

## 2. Related Works

Palmprint and hand geometry are two well-explored biometric characteristics. Palmprint is characterized by a rich texture consisting of ridges, singular and minutia points. This texture can be extracted from high-resolution images that are suitable for forensic applications such as criminal detection [32]; for civil and commercial applications, low-resolution images are employed; in this case, only principal lines and wrinkles are analyzed [33,34]. A wide variety of feature extraction methods have been proposed including line-based, texture-based, and appearance-based features [33,34,35]. More recently, several holistic and coding techniques were evaluated as well as machine/deep learning approaches [36,37,38]. Most palmprint recognition systems use 2-D images for feature extraction, increasingly collected in a contactless way. However, 2-D palmprint images can be easily counterfeited. To overcome such difficulties, optical palmprint recognition systems that use 3-D information on the curvature of the palm were proposed [39].

Biometric systems based on hand geometry, which use a varying number of distances including lengths and/or widths of palm and fingers as templates, have been under development since the second half of the 20th century.The main technology used for capturing an image of the human hand is the optical approach but good recognition results were obtained using infrared radiation as well [8,40,41]. A number of pegs is sometimes used to obtain correct alignment of the fingers; however, unconstrained acquisition modality is nowadays the one most commonly adopted.

Both palmprint and hand geometry were extracted from volumetric ultrasound images.

Various probes and ultrasonic scanners were used for collecting a volumetric region of the palm in a reasonable time period (about 5 s) and with a resolution better than 100 dpi [42,43,44]. In total, two types of 3D palmprint features were extracted. The first feature was based on palm curvature [25], using similar methods as those used for optical images. The second feature was based on the analysis of principal lines extracted from several depths under the skin [27], gaining 3D information that cannnot be achieved with any other technology.

Three-dimensional ultrasonic images of the whole hand were acquired with a similar technique as that used for palmprint [23]. To minimize acquisition time, as the volume to be collected was much greater, the 3D hand images were quite lower in resolution. Furthermore, in this case several 2D images were extracted from various depths under the skin. For each image, a template based on a number of distances was defined and all these templates were opportunely combined to provide a 3D template [24].

Biometric fusion can be performed at several levels including at the sensor level, feature level, score level, and decision level.

Sensor-level fusion mainly consists of fusing raw samples of biometric traits acquired by the sensor [45]. This fusion technique can be performed if the samples are compatible and represent the same biometric trait. Moreover, it is mostly employed in multi-sample systems where multiple samples are combined to obtain a composite sample for human identification.

Feature-level fusion combines feature vectors obtained during the feature-extraction phase. This technique can be applied when feature sets of different modalities are compatible or synchronised [46].

Score-level fusion is based on a combination of match score levels by using the resultant score, in order to perform a final recognition decision [47].

Decision-level fusion is similar to score-level fusion with the difference that scores are turned into match/non-match decisions before fusion [48,49].

Among the above-described methods, fusion at the score level is the most popular because it is easy to implement and allows adequate information content [50]. Over the years, a large number of score-level fusion algorithms have been experimented with. Hammandlu et al. [47], Peng et al. [51], and El-latif et al. [52] proposed fusion approaches based on t-norms. Many authors have proposed score-level fusion methods based on the weighted score sum rule where the appropriate weight is assigned to each score. Weights were calculated through different modalities: for instance, Zhang et al. defined the weight on the basis of the EER of each modality [53]; Damer et al. estimated the weights through the mean of scores distribution and its maxima [54]; Kabir et al. calculated the weight through distances between max and mean or mean and min of genuine/impostor scores [55]; Poh et al. and Snelick et al. defined the weight on the basis of mean and standard deviation of genuine and impostor scores [56,57].

## 3. Image Acquisition and Feature Extraction

Ultrasound image acquisition of the human hand [24] is performed with a system composed of an ultrasound scanner [58], a linear array of 192 elements and a numerical pantograph, which controls the movement of the probe on the region of interest (ROI).

The acoustic coupling between the human body and the probe is created by submerging both in a tank of water. A three-dimensional image is acquired by moving the probe along the elevation direction; during the motion, several B-mode images are collected and regrouped in order to obtain a volume defined by an 8-bit grayscale 3D matrix (416 × 500 × 68 voxels). Figure 1 shows an example of a 3D render of the whole human hand. The resolution of the image is about 400 μm.

Successively, an interpolation is performed along the z-axis and 2D renderings are extracted at various depths from the volume: the external surface of the hand is first projected on the XY plane in order to achieve the shallowest 2D image. Then, it is translated along the z-axis beneath the skin and projected again on the XY plane obtaining 2D images at increasing depths.

Three-dimensional information is taken into account by collecting fourteen 2D images with a step of 50 μm: the shallowest image is taken at 100 µm while the deepest is captured at 750 µm. Successively, 2D and 3D features both for hand geometry and palmprint are extracted from 2D renderings: for the hand geometry, they consist of hand measurements including the size of palm, lengths and widths of fingers, while for palmprint they are represented by principal lines and main wrinkles.

The procedure employed for the extraction of 2D templates consists of a median filter to reduce the noise, binarization with a suitable threshold, and calculation of distances between a middle point on the wrist boundary, as a reference point, and each point on the hand contour with Euclidean distance [24]. Then, several feature points, shown in Figure 2a with different colours, including finger peaks, a middle point, valleys between fingers, other finger base points, and an extra point are extracted [40]. From these points, 26 distances are calculated in order to define a 2D template. Successively, 2D templates at different depths are combined to obtain the 3D template; three combinations were considered:**Mean features (MF)**: each length computed as the mean value of the lengths obtained at each depth;**Weighted Mean features (WMF)**: each length represented by a weighted mean of the lengths obtained at various depths;**Global features (GF)**: all lengths computed at every depth.

Regarding palmprint, a palm ROI is first extracted by defining a square as indicated in Figure 2b; in this way, the repeatability of the procedure is guaranteed. Successively, after some preprocessing operations, 2D features are extracted with a classical line-based procedure [27] as shown in Figure 3. The image is scanned along four directions (0°, 90°, 180°, 270°). Along each direction, the edges of principal lines are detected by calculating intensity variations through the first derivative. Short lines and isolated points are then filtered by using a Laplacian filter. The four images obtained are summed with a logical OR operation. Finally, morphological operations are executed: closing operation in order to filling holes and small concavity, thinning operation, and pruning operation to remove short lines. In this way, 2D templates are achieved.

Successively, 2D templates, at different depths, are combined with a particular algorithm in order to obtain a 3D template. The algorithm is mainly based on two operations that are executed iteratively: the first analysed 2D template $T_{i}$ is dilated with a structuring element of β dimension and stored in a 3D matrix; then, a logical AND comparison is performed between the current dilated template and the 2D template at adjacent depth level ($T_{i-1}$ or $T_{i+1}$) [31,59], and the result is stored in the 3D matrix. The dilation operation is performed to account for the fact that, by increasing the under-skin depth, the palm trait may be not orthogonal to the XY plane, while the AND operation allows filtering spurious traits in each of the two images. Dimension β of the structuring element affects the quality of results because if it is too high, the 3D template may contain spurious traits while if it is too low, some principal information could be eliminated.

Figure 4a shows an example of a 3D template represented as a colour scale matrix and obtained by setting β = 5, where each pixel is defined by a value that varies from 0 to 13: 0 defines a blue pixel that corresponds to the absence of trait while 13 defines a dark red pixel that corresponds to the presence of the trait at all depths. For comparison, Figure 4 shows the corresponding 2D grey scale render.

## 4. Fusion

In a previous work [31], a multimodal system based on 3D hand geometry and 3D palmprint was investigated where it was assumed that the best fusion results were achieved by considering 3D templates, both for hand geometry and palmprint, that provided the best recognition results in unimodal experiments. Instead, in the present work, this assumption has been removed and extensive fusion experiments have been performed by testing several fusion methods based on the weighted score sum rule and by considering a wide variety of possible combinations of palmprint and hand-geometry templates.

### Experimented Weighted Score Sum Rules

Methods based on the weighted score sum rule are easy to implement and demonstrate high effectiveness [50]. They are generically expressed through the following equation:(1)$$R_{MW}=\sum_{i=1}^{n}w_{i}R_{i}$$
where *n* is the number of characteristics, $R_{i}$ represents the score and $w_{i}$ is the corresponding weight. Several kinds of weights were experimented with. In the following, only those discussed in Section 2 are evaluated. In all cases, the weight is calculated according to the expression:(2)$$w_{i}=\frac{y_{i}}{\sum_{j=1}^{n}y_{j}}$$

For each methodology, the value $y_{i}$ is defined with a certain modality:**EER weighted (EERW)** [31,53]:
(3)$$y_{i}=\frac{1}{EER_{i}}$$**D-Prime weighted** [57]:
(4)$$y_{i}=\frac{\mu_{i}^{G}-\mu_{i}^{I}}{\sqrt{\sigma_{i}^{G^{2}}+\sigma_{i}^{I^{2}}}}$$**Mean-to-extrema weighted** [54]:
(5)$$y_{i}=\left(Max_{i}^{I}-\mu_{i}^{I}\right)+\left(\mu_{i}^{G}-Max_{i}^{G}\right)$$**Fisher’s discriminant ratio weighted (FDRW)** [56]:
(6)$$y_{i}=\frac{{\left(\mu_{i}^{G}-\mu_{i}^{I}\right)}^{2}}{\sigma_{i}^{G^{2}}+\sigma_{i}^{I^{2}}}$$**Kabir method** [55]:
(7)$$y_{i}=\left(Max_{i}^{G}-\mu_{i}^{G}\right)+\left(\mu_{i}^{I}-Min_{i}^{I}\right)$$
where $\mu_{i}^{G}$ and $\mu_{i}^{I}$ are the means of genuine and impostor scores distributions, respectively, $\sigma_{i}^{G}$ and $\sigma_{i}^{I}$ are the standard deviations of genuine and impostor scores distributions, respectively, $Max_{i}^{G}$ and $Max_{i}^{I}$ are the maximum values of genuine and impostor scores, respectively, and $Min_{i}^{I}$ is the minimum value of impostor scores.

## 5. Results

Recognition accuracy is evaluated by performing verification and identification experiments on a database previously employed in [31]. It is composed of 110 samples acquired from 50 different users of both sexes with ages ranging from 18 to 55.

### 5.1. Verification

The verification mode consists of identifying a person through their claimed identity and is based on one-to-one comparisons between a query template and a reference template that is stored in the database. Verification experiments are performed by comparing each 3D template with all others in the database both for palmprint and hand geometry. Regarding hand geometry [24,31], 3D templates are compared by employing absolute distance function:(8)$$D=\sum_{i=1}^{n}\left|Q_{i}-R_{i}\right|$$
where $Q_{i}$ is the query template while $R_{i}$ is the reference template. Instead, the similarity criterion between two 3D palmprint templates [27,59], of the type shown in Figure 4a, is defined by a classic pixel-to-area approach based on a logical AND operation between corresponding pixels of two images:(9)$$S_{3D}\left(R,Q\right)=\frac{2}{S_{R}+S_{Q}}\sum_{i=1}^{n}\sum_{j=1}^{n}T_{R}\left(i,j\right)⊕T_{Q}\left(i,j\right)⊕|O_{R}\left(i,j\right)-O_{Q}\left(i,j\right)|<\alpha$$
where $T_{R}$ and $T_{Q}$ are the reference and query templates, respectively, n×n is the dimension of the template and $S_{R}$ and $S_{Q}$ are the sum of pixels of value “1” in $T_{R}$ and $T_{Q}$, respectively; α is an integer value between 0 and the number of 2D templates and acts as a filter for small or secondary traits. The lower the value of α, the greater the filter effect.The term |$O_{R}\left(i,j\right)-O_{Q}\left(i,j\right)$|<α allows tuning the acceptable difference of occurrences in corresponding pixels.

The result of the comparison, referred to as score, is defined as genuine or impostor if the two templates come from the same user or from different users, respectively. Furthermore, if the score exceeds a certain threshold, the user is authenticated, while if the score is lower, the user is rejected. In biometric systems, two types of errors may exist: false acceptance and false rejection errors, which occur when an impostor score exceeds the threshold and when a genuine score is lower than the threshold, respectively. Consequently, the performance of a system is evaluated by the False Acceptance Rate (FAR) and the False Rejection Rate (FRR), which can be computed as the ratio between occurrences of false acceptances and false rejections and the total scores, respectively. The Equal Error Rate (EER), which occurs when FRR = FAR, is often used to provide a synthetic evaluation of the recognition capability of the system. Performances of different systems are compared through Detection Error Tradeoff (DET) and Receiver Operating Characteristics (ROC) curves, which plot FRR and True Acceptance Rate (TAR), defined as 1-FRR, as a function of FAR, respectively. For DET and ROC curves, the system shows better recognition performances when the corresponding curve is closer to the axis. Specifically, in order to quantitatively evaluate the performances on the basis of ROC curves, the Area Under Curve (AUC) is computed.

Figure 5 shows DET and ROC curves obtained for palmprint by varying α values between 3 and 9, for β = 3, β = 4, and β = 5. The values of EER and AUC computed for each curve of Figure 5 are reported in Table 1. As can be seen, an overall improvement of recognition results is observed by increasing β. This improvement strictly occurs for the AUC. As concerns EER, best values are obtained with β = 5 for any value of α, while, in most cases, β = 3 provided better results than β = 4. It is also to note that, by increasing β, lowest EER values are achieved for decreasing values of α, i.e., α = 7 and α = 8 for β = 3, α = 6 and α = 7, for β = 4, and α = 5 and α = 6 for β = 5. A similar behaviour can be observed for AUC and indicates that a higher dilation (β) before the AND operation in 3D template generation should be compensated by accepting only small differences in pixel occurrences between the two 3D templates during the matching operation. EER and AUC values obtained by using different 3D templates for hand geometry are reported in Table 2. A detailed comparison between DET curves was reported in a previous work [24]. Successively, the two characteristics are fused by employing the methods described in Equations (Equation 1)–(Equation 7) and by considering all analyzed combinations of α and β for palmprint while, for hand geometry, only the 3D template obtained with GF is used in fusion operations. Fusion results are reported in Table 3, Table 4 and Table 5 for β = 3, β = 4, and β = 5, respectively. As can be seen, the fusion between hand geometry and palmprint allows obtaining, in most cases, an improvement in the recognition capabilities in terms of EER. In particular, a notable lowering of the EER, with respect to that of the best unimodal methods, is found for the great majority of cases for β = 5; the lowest values (about 0.06%) were achieved with the EERW (α = 4 and α = 5) and Kabir (α = 4, α = 5, and α = 8) methods. The value of 0.074%, obtained for β = 3 and α = 9, is noteworthy as well. It is to highlight that, for all the above-reported cases, the fused EER is lower than the one obtained by choosing β = 5 and α = 6, which provided the best result for the unimodal characteristic [31]. The worst fusion results were instead obtained for β = 3, in particular with FDRW and D-Prime methods, for which an increase of EER is found.

Figure 6 shows DET curves plotted for the best fusion methods and for best unimodal palmprint and hand-geometry cases, where the dramatic improvement in fusion recognition results over the unimodal results can be observed. The first bisector FRR = FAR is not plotted for figure readability. As far as the AUC values are concerned, all fusion methods demonstrate an improvement. In particular, for β = 5, EERW, D-Prime and Kabir methods allow obtaining an AUC value equal to 100% for all α values; the same result is achieved for α = 9 β = 4 as well.

### 5.2. Identification

Identification is an alternative modality to verification with the purpose of assigning an identity to an unknown person. The system compares a test template with all templates contained in a database: the highest score determines the identity if it exceeds a predefined threshold; otherwise, the person is considered as not present in the database.

Identification experiments were performed only for fusion scores that provided the best results in verification experiments, i.e., those obtained for β = 5 with the Kabir method (α = 4, α = 5, α = 8), EERW (α = 4, α = 5) and for β = 4 with D-Prime (α = 7), and for the best palmprint and hand geometry cases.

Matching results are stored in 110 tables, one for each sample, where each one contains 109 scores that follow a descending order. An identification experiment is successful when all genuine scores fill the first positions of the table, i.e., the lowest genuine is higher than the highest impostor score. The identification rate is defined by the number of success tables over the total number of tables. An identification rate of 100% is registered for all analysed fusion methods, while the best unimodal palmprint and hand geometry methods scored identification rates of 98% and 91%, respectively, again demonstrating the effectiveness of fusion.

To further test the robustness of the identification procedure, normalized score differences (NSD) between the lowest genuine and highest impostor scores were calculated for all experiments. Figure 7 shows the distributions of such values normalized to the lowest genuine score for the six fusion methods and the best palmprint and hand geometry cases. For each distribution, mean, standard deviation, and the number of occurrences when NSD is lower than 0.1 are calculated and reported in Table 6. As can be seen, fusion based on D-prime seems to be the most robust because it exhibits the highest mean value and the lowest number of NDS < 0.1 among the fusion methods; the highest value of standard deviation is observed as well. Note that a strict correlation between the three parameters seems to occur for any fusion method. In fact, a decrease in the occurrences of NSD < 0.1 corresponds to an increase in both mean and standard deviation. For hand geometry and palmprint, mean and standard deviation are reported for comparison while, as they exhibited an identification rate lower than 100%, the occurrences of NSD < 0.1 are not reported at all.

## 6. Conclusions

In this work, recognition performances of a multimodal system based on the fusion of three-dimensional palmprint and hand-geometry features, extracted from the same volumetric ultrasound images, are experimentally evaluated. Several fusion techniques based on the weighted score sum rule and on a wide variety of possible combinations of palmprint and hand geometry scores, obtained by varying two main parameters (α and β) that sensibly affect palmprint score results, are proposed and tested. Recognition capabilities of the various methods are evaluated and compared by carrying out verification and identification experiments on a homemade database employed in a previous work [31]. Verification results demonstrated that the fusion, in most cases, allows obtaining a dramatic improvement in recognition performance with respect to unimodal systems. Particularly, an EER value of about 0.06% is achieved in at least five cases against values of 1.18% and 0.63% of the best cases for unimodal palmprint and hand-geometry, respectively. Moreover, it was also proved that the best fusion results are not obtained by fusing the best scores of the two unimodal characteristics. Identification experiments, which are executed for the fusion methods that provided the best verification results, demonstrated an identification rate of 100%, against 98% and 91% obtained in the best cases for unimodal palmprint and hand-geometry, respectively, again demonstrating the effectiveness of fusion.

The exceptionally high-recognition accuracy, together with the other features of ultrasound (above all the capability of effectively detecting liveness), makes this kind of system particularly suited for high secure access applications.

Future work will be devoted to experimenting with the acquisition of volumetric ultrasound images of the hand by employing gel as a coupling medium, instead of water, as already tested in previous works [59,60]. The benefits offered by this coupling approach include reduced invasiveness of the acquisition procedure and increased user comfort regarding hand placement. This approach will make it easier to establish a wider database, which will improve the effectiveness and reliability of the achieved results. In addition to the extraction of hand geometry and palmprint, the coupling approach will also allow vein patterns to be extracted from the same collected volume. Finally, alternative feature-extraction methods, mainly based on machine learning and deep learning [61], will be investigated in addition to other fusion techniques, particularly feature-level techniques.

## Figures and Tables

**Figure 1 sensors-23-03653-f001:**
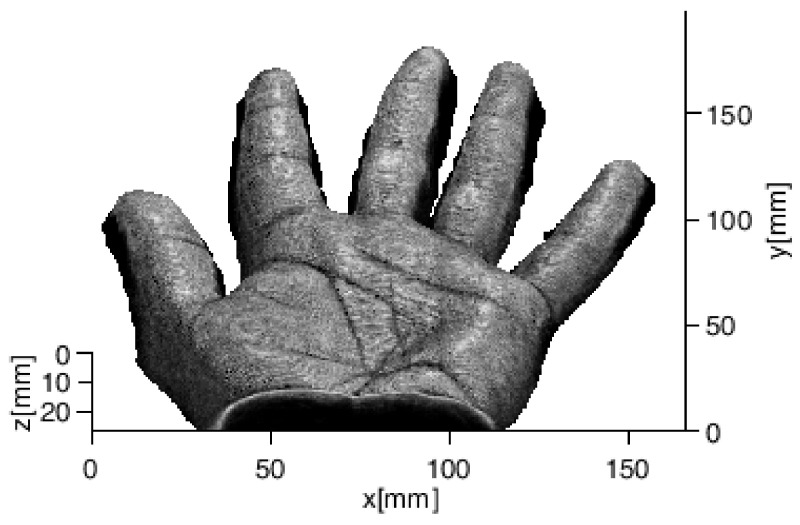
Example of 3D rendered human hand.

**Figure 2 sensors-23-03653-f002:**
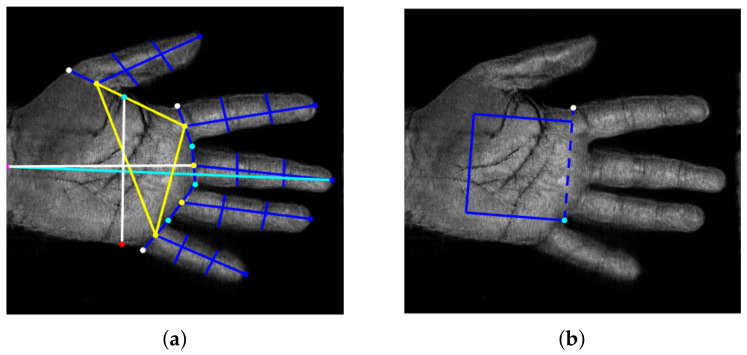
(**a**) Feature points extracted from the hand shape and the 26 distances defining the 2D template; (**b**) ROI extraction for palmprint from two feature points of (**a**).

**Figure 3 sensors-23-03653-f003:**
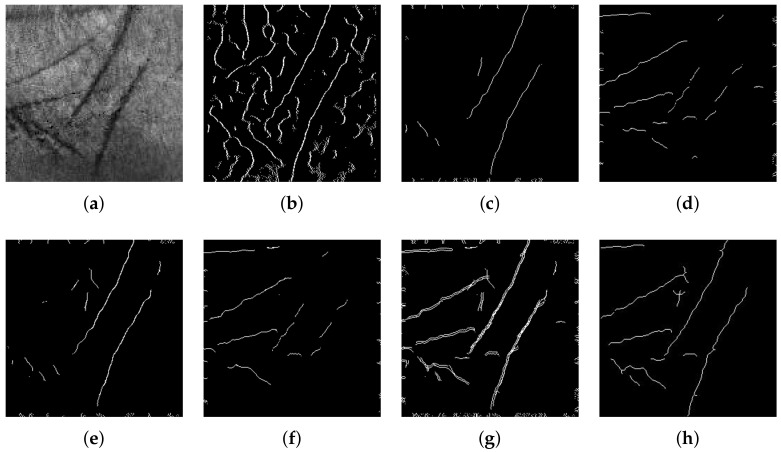
Palmprint feature extraction procedure step by step: (**a**) 2D grayscale image at 350 μm; (**b**) image after detection of edges; (**c**) feature extraction along direction 0° (**d**) 90° (**e**) 180° (**f**) 270° (**g**); logical OR of images after feature extraction along four directions; (**h**) final 2D template.

**Figure 4 sensors-23-03653-f004:**
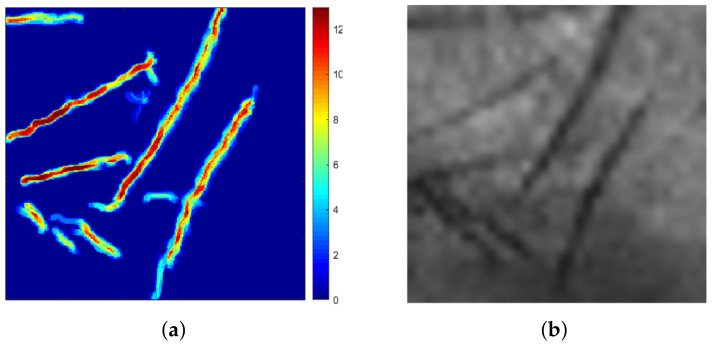
(**a**) Three-dimensional template represented as a colour scale matrix where the trait’s depth varies from 0 to 13; (**b**) 2D greyscale render of the same sample.

**Figure 5 sensors-23-03653-f005:**
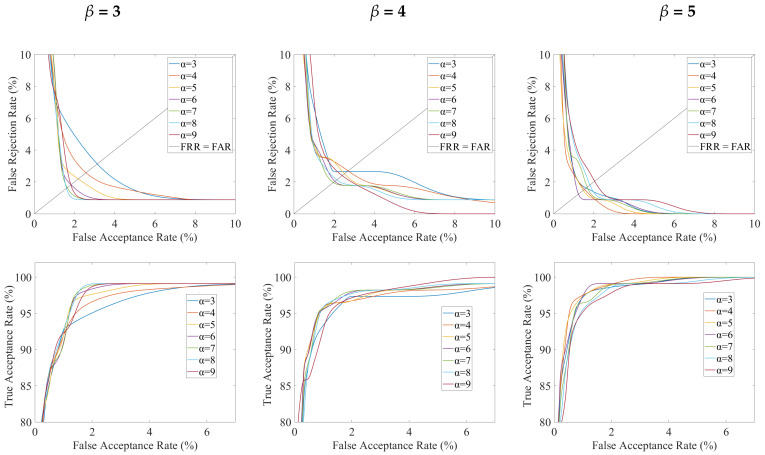
DET (first line) and ROC (second line) curves obtained with palmprint templates by varying α and β values.

**Figure 6 sensors-23-03653-f006:**
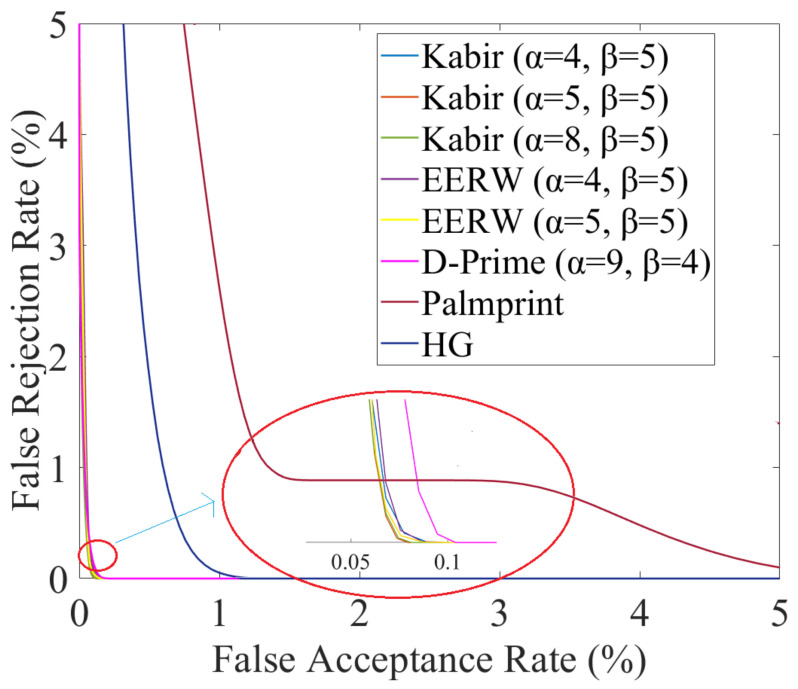
DET curves obtained with the best fusion methods. Best unimodal palmprint and hand geometry curves are also reported for comparison.

**Figure 7 sensors-23-03653-f007:**
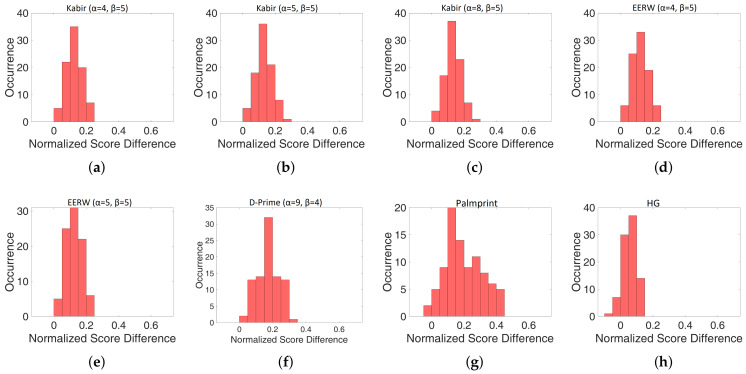
Distribution of Normalized Score Difference (NSD) between the highest impostor score and the lowest genuine score for: (**a**) Kabir (α = 4, β = 5), (**b**) Kabir (α = 5, β = 5), (**c**) Kabir (α = 8, β = 5), (**d**) EERW (α = 4, β = 5), (**e**) EERW (α = 5, β = 5), (**f**) D-Prime (α = 9, β = 4), (**g**) Palmprint, (**h**) HG. As can be seen, for fusion methods, NSD values are always higher than 0, ensuring an identification rate equal to 100%.

**Table 1 sensors-23-03653-t001:** 3D Palmprint: EER and AUC values for all curves reported in Figure 5.

	β=3		β=4		β=5	
Method	EER	AUC	EER	AUC	EER	AUC
α=3	3.08%	99.35%	2.00%	99.61%	1.60%	99.85%
α=4	2.55%	99.43%	2.75%	99.69%	1.54%	99.88%
α=5	2.13%	99.50%	2.53%	99.71%	1.48%	99.88%
α=6	1.82%	99.53%	1.93%	99.70%	1.18%	99.87%
α=7	1.59%	99.53%	1.91%	99.69%	1.64%	99.84%
α=8	1.54%	99.50%	2.08%	99.67%	1.78%	99.82%
α=9	1.75%	99.55%	2.49%	99.77%	2.04%	99.79%

**Table 2 sensors-23-03653-t002:** 3D Hand Geometry: EER and AUC values for the three types of 3D templates.

Method	EER	AUC
GF	0.64%	99.94%
MF	0.74%	99.94%
WMF	0.93%	99.94%

**Table 3 sensors-23-03653-t003:** EER and AUC values obtained using various fusion methods and α values for β = 3.

	ERRW		D-Prime		FDRW		MEW		Kabir	
Method	EER	AUC	EER	AUC	EER	AUC	EER	AUC	EER	AUC
α=3	0.22%	99.99%	0.90%	99.98%	0.89%	99.98%	0.20%	99.99%	0.24%	99.99%
α=4	0.24%	99.99%	0.88%	99.98%	0.90%	99.97%	0.26%	99.99%	0.28%	99.99%
α=5	0.28%	99.99%	0.86%	99.98%	0.90%	99.97%	0.33%	99.99%	0.38%	99.99%
α=6	0.30%	99.99%	0.90%	99.98%	0.90%	99.96%	0.22%	100%	0.37%	100%
α=7	0.33%	99.99%	0.88%	99.98%	0.89%	99.96%	0.23%	100%	0.36%	100%
α=8	0.32%	99.99%	0.90%	99.97%	0.80%	99.97%	0.24%	100%	0.37%	100%
α=9	0.34%	100%	0.90%	99.98%	0.90%	99.96%	0.33%	99.99%	0.63%	99.99%

**Table 4 sensors-23-03653-t004:** EER and AUC values obtained using various fusion methods and α values for β = 4.

	ERRW		D-Prime		FDRW		MEW		Kabir	
Method	EER	AUC	EER	AUC	EER	AUC	EER	AUC	EER	AUC
α=3	0.21%	99.99%	0.55%	99.99%	0.81%	99.99%	0.25%	99.98%	0.14%	99.99%
α=4	0.16%	99.99%	0.52%	99.99%	0.85%	99.99%	0.21%	99.99%	0.18%	100%
α=5	0.16%	99.99%	0.68%	99.99%	0.86%	99.99%	0.16%	99.99%	0.16%	100%
α=6	0.14%	100%	0.63%	99.99%	0.90%	99.98%	0.094%	99.99%	0.90%	100%
α=7	0.47%	99.99%	0.47%	99.99%	0.65%	99.99%	0.47%	100%	0.20%	100%
α=8	0.15%	100%	0.75%	99.99%	0.90%	99.99%	0.14%	100%	0.21%	100%
α=9	0.16%	100%	0.074%	100%	0.15%	100%	0.15%	100%	0.14%	100%

**Table 5 sensors-23-03653-t005:** EER and AUC values obtained using various fusion methods and α values for β = 5.

	ERRW		D-Prime		FDRW		MEW		Kabir	
Method	EER	AUC	EER	AUC	EER	AUC	EER	AUC	EER	AUC
α=3	0.14%	100%	0.20%	100%	0.14%	100%	0.27%	100%	0.33%	100%
α=4	0.063%	100%	0.12%	100%	0.29%	100%	0.18%	100%	0.058%	100%
α=5	0.063%	100%	0.22%	100%	0.28%	99.99%	0.12%	99.99%	0.062%	100%
α=6	0.081%	100%	0.22%	100%	0.24%	99.99%	0.41%	99.99%	0.098%	100%
α=7	0.15%	100%	0.20%	100%	0.30%	99.99%	0.15%	100%	0.1%	100%
α=8	0.13%	100%	0.23%	100%	0.27%	100%	0.15%	100%	0.06%	100%
α=9	0.15%	100%	0.24%	100%	0.29%	99.99%	0.083%	100%	0.088%	100%

**Table 6 sensors-23-03653-t006:** Mean, Standard Deviation and occurrences of NSD < 0.1 for the distributions of Figure 7.

Method	Mean	Standard	NSD < 0.1
		Deviation	
Kabir (α = 4, β = 5)	0.1255	0.0498	27
Kabir (α = 5, β = 5)	0.1313	0.0501	23
Kabir (α = 8, β = 5)	0.1328	0.0503	21
D-Prime (α = 7, β = 4)	0.1785	0.0655	15
EERW (α = 4, β = 5)	0.1209	0.0488	31
EERW (α = 5, β = 5)	0.1247	0.0487	30
HG	0.0585	0.0423	
Palmprint	0.2085	0.1251

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
