# Peer review of "Recognition Performance Analysis of a Multimodal Biometric System Based on the Fusion of 3D Ultrasound Hand-Geometry and Palmprint"

_sensors, 2023, doi:10.3390/s23073653_

Round 1

Reviewer 1 Report

This manuscript extensively evaluated the performance of several multimodal recognition systems obtained by fusing the features of palmprint and hand-geometry, which are extracted from the same 3D image of a palm. Verification results demonstrated that in most cases, the fusion obtains a noticeable improvement compared to single modal system.

Firstly, the novelty of this work is not ample enough, as it only compared the performance of several existed fusing algorithms on recognition, without proposing any novel algorithm or finding a best algorithm to apply to multimodal recognition. Secondly, this paper cites the reference [50] too many times, so that some detailed information is lost, like the definition of the network used, and the definition of α and β is not clear enough. Last, the Introduction part has only one single paragraph, including background, related research and chapter arrangement.

Main concerns:

1. There is a lack of innovation in the method of this paper. Much of the innovation is to propose a new algorithm or choose a best algorithm and demonstrate it, but there is too little detail to provide sufficient guidance or inspiration. The conclusion of optimization is not obvious enough.

2. Much of the literature on palmprint recognition has been published in international journals, i.e. IEEE TIP, TIFS, TIM, TCSVT in recent years. It is a pity that the author did not pay attention to the state of the art.

3. Although the point of this work is the application and optimization of features fusing on multimodal recognition, a brief introduction of the detail of the network design is recommended. Also a more clear definition of α and β is recommended.

4. The only paragraph in chapter 1 is too long, with the content of background, related research and chapter arrangement. It’s better to divide it into several paragraphs.

5. There are some problems about grammar. Like “sexs” in line 192, “sexes” is correct. And for “aapplications” and “particulary” in line 303, “applications” and “particularly” is correct.

Reviewer 2 Report

The paper is about a multimodal biometric system using hand-geometry and palmprint. The paper is well described and addresses major key points in this field. But the contribution novelty is limited. Hand-geometry and palmprint features have been well explored in the past few years and thus the idea of fusion is not very new. However, the overall work technically sounds good. 

There are grammatical corrections needed throughout the paper. Some sentences are confusing/unclear such as "...combinations of palmprint and hand geometry scores, achieved by varying two main parameters,..." 

Some typos: "...weighted (FDRW)[66]:..."

From table 3, it is observed that alpha is not constantly performing for all datasets when beta is set to 3. While when beta is 4, alpha 9 is constantly performing for all datasets with highest AUC. 

What is the exact feature extraction method used in this paper? Section 3 is unclear.

Too many references. 

Reviewer 3 Report

Author must represents the work with Flow Chart and Step by step explanation required. 

Novelty is missing in the entire manuscript.

Literature section is missing.

Author must use 2 standard database for validation of the proposed work.

Round 2

Reviewer 1 Report

This manuscript extensively evaluated the performance of several multimodal recognition systems obtained by fusing the features of palmprint and hand-geometry, which are extracted from the same 3D image of a palm. Verification results demonstrated that in most cases, the fusion obtains a noticeable improvement compared to single modal system.

After revising, the main concerns has been appropriately responded. But there is still a tiny ambiguity problem in chapter 2 which may cause logically incoherence.

Main concerns:

1. The sentence in line 93 and 94 shows that both two biometrics have been extracted by ultrasound images which doesn’t show the convenience and property of ultrasound images to research in multimodal recognition obtained by fusing the features of 3D palmprint and hand-geometry, to lead to the following paragraph.

Reviewer 2 Report

The previous comments are addressed and the paper is now improved to a great extent. The paper is application wise new  and can be acceptable. 

Author Response

Thank you